theoretical biology/evolution/mathematical modelling

mating types, self-incompatibility, balancing selection, isogamy, negative frequency-dependent selection

# Fitness differences suppress the number of mating types in evolving isogamous species

Yvonne Krumbeck[1], George W. A. Constable[2] and Tim Rogers[1]

[1]Department of Mathematical Sciences, University of Bath, Bath BA2 7AY, UK
[2]Department of Mathematics, University of York, York YO10 5DD, UK

GWAC, 0000-0001-9791-9571; TR, 0000-0002-5733-1658

Sexual reproduction is not always synonymous with the existence of two morphologically different sexes; isogamous species produce sex cells of equal size, typically falling into multiple distinct self-incompatible classes, termed mating types. A long-standing open question in evolutionary biology is: what governs the number of these mating types across species? Simple theoretical arguments imply an advantage to rare types, suggesting the number of types should grow consistently; however, empirical observations are very different. While some isogamous species exhibit thousands of mating types, such species are exceedingly rare, and most have fewer than 10. In this paper, we present a mathematical analysis to quantify the role of fitness variation—characterized by different mortality rates—in determining the number of mating types emerging in simple evolutionary models. We predict that the number of mating types decreases as the variance of mortality increases.

## 1. Introduction

Reproductive processes play a key role in evolutionary biology. Yet the evolution of sexual reproduction itself is still an intensely debated subject that gives rise to many puzzling questions [1–3]. For a complete understanding, it is important to consider biological features that, although sometimes taken for granted as common, actually display startling variety across the tree of life [4]. In this paper, we will be concerned with just such a feature and its evolution; the number of 'sexes' in a given species.

For clarity, let us compare the more familiar sexual features of mammals with those of the slime mould *Dictyostelium discoideum*.

Mammals possess two sexes, defined in terms of the size of the gametes (sex cells) that they produce; males produce small, motile and numerous sperm while females produce large, sessile and well-provisioned eggs. A union between haploid gametes of opposite types (sperm and egg) permits the formation of a zygote. This state, in which gametes display a clear size dimorphism, is termed anisogamy. By contrast, *D. discoideum* is an isogamous species; it produces gametes that are morphologically indistinguishable [3]. These gametes still come in a number of self-incompatible, genetically determined variants, termed mating types, that can be understood as ancestral analogues of the sexes. However, unlike true sexes, the number of mating types is not restricted to two; *D. discoideum* has three mating types [5].

A natural question that then arises is what drives the evolution of the number of mating types? Within anisogamous species, a series of trade-offs (it is difficult to produce gametes that are both small and well-provisioned or large and numerous) restrict the number of morphological types to two [6]. By contrast, isogamous species, in which such trade-offs are absent, do not face such a restriction. In fact, simple evolutionary reasoning suggests that a population with two mating types should be a very unstable configuration.

To explain this idea, let us discuss the following scenario: an isogamous population with two distinct self-incompatible mating types *A* and *B* of equal frequencies. Assuming mass-action encounter rates, individuals of each type have a 50% chance of locating an individual of the opposite complementary type within the population. Now we introduce a novel self-incompatible mating type *C* at low frequency. An individual of this rare type *C* is now able to mate with all individuals of type *A* and *B* (a large proportion of the population) and thus is selected for, until the population reaches a new equilibrium in which all three types have equal frequencies. This 'rare sex advantage' to novel isogamous mating types leads to the prediction that their number should consistently grow [7]. In a very extreme case, we might imagine as many distinct mating types as there are individuals in the population.

By the above theoretical argument, one might predict that *D. discoideum*, with its three mating types, should be an evolutionary outlier, restricting its opportunities for sexual reproduction to a mere two-thirds of the population. In fact, the empirical data paints rather the opposite picture. The majority of isogamous species (including the yeast *Saccharomyces cerevisiae* [8], the ciliate *Blepharisma japonicum* [9] and the green algae *Chlamydomonas reinhardtii* [10]) have two mating types [11]. A smaller variety of species have a handful of mating types (including the fungi *Mycocalia denudata* and *Agaricus bisporus*, and the ciliates *Tetrahymena hyperangularis* and *Euplotes raikovi* with 9, 18, 4 and 12 mating types, respectively [9,12]). Exceedingly few have many types (the mushroom fungus *Coprinellus disseminatus* has 143 mating types [13], while *Schizophyllum commune* has an astounding 23 328 different mating types [14]).

Multiple biologically complex theories have been proposed to correct this discrepancy between model prediction and observation (reviewed in [15]). One hypothesis is that the complementary receptor or pheromone signalling system of a population with two mating types restricts the emergence of new types [16] (e.g. a third type, derived from the residents, will not be as well adapted to the extant types as they are to each other, and thus will be selected against [17]). Another theory contends that where gametes mix their cytoplasm, the evolution of uniparental inheritance of organelles (e.g. mitochondria) to prevent cytoplasmic conflicts consequently selects for a single cytoplasmic donor and receiver [18] (more complicated donor–receiver relationships of multi-mating type systems become increasingly unstable [19]). A third hypothesis states that the dynamics for the mating type encounter rate may limit the selection for rare mating types [7] (if gametes experience no mortality pressure, they can survive until a compatible partner becomes available, eliminating selection for more than two mating types).

While each of these hypotheses may be biologically plausible, they also lead to a somewhat bimodal prediction for the number of mating types; under certain conditions two (or sometimes three) mating types are selected for, while outside these conditions the number of mating types should consistently grow. Thus these theories do not account for the intermediate number of mating types found in ciliates and fungi.

Recently, attention has been turned to the power of neutral models for an explanation as to the number of mating types observed across species [20–22]. These models take a similar approach to [7] in assuming that there are no differences (other than self-incompatibility) between types. Their analysis considers population genetic effects of finite-population size and investigates the number of mating types predicted at long times under a mutation–extinction balance. Conceptually, this approach is similar to classic population genetic studies estimating the number of self-incompatibility

alleles in certain plant populations [23–25]. (In these species, further self-incompatibilities have arisen between the sexes themselves in order to prevent inbreeding depression [26,27].)

A key biological feature affecting the typical dynamics of the self-incompatibility alleles that define mating types is the balance between sexual and asexual reproduction. Many isogamous species are facultatively sexual, experiencing only rare bouts of sexual reproduction between many rounds of asexual division. In [28], it was shown that prolonged rounds of asexual division could substantially increase the extinction risk to mating type alleles. In [20], it was then shown analytically that the expected number of mating types under a mutation–extinction balance would decrease dramatically as a function of decreasing rates of sex. These results were extended in [22] to predict the invasion and extinction dynamics of mating type alleles in finite populations, showing that these predictions were broadly consistent with empirical observations.

The above neutral models each assume that every mating type experiences a symmetric negative frequency-dependent selection (the rare sex advantage), but that otherwise each mating type is equally fit. However, as verbally argued in [29], such perfect symmetry in fitnesses may be unlikely. In many diverse lineages, mating type loci are found to feature large regions of suppressed recombination [10,30], analogous to non-recombining sex chromosomes in animals. This suppressed recombination has two important consequences. First, linked deleterious mutations are more likely to accumulate on mating type alleles as a function of their reduced effective population size [30,31]. Second, we expect that suppressed recombination will rapidly lead to divergence between the mating type alleles [32].

In obligately sexual species with just two mating types, any effect of fitness differences between mating type alleles would be somewhat muted by Fisher's principle [33]; the need to reproduce sexually guarantees an equal abundance of each mating type in a pool of successfully partnered parents. However, when sex is facultative, differences in the reproductive rate of mating types during asexual reproduction can distort the population's sex-ratios away from an even distribution [34]. In an extreme scenario then, fitness differences between the mating types could drive competitive exclusion and a reduction in mating type diversity. Preliminary simulations in [20] supported this view, but no analytic results were obtained.

In this paper, we will address this gap in the literature, focusing on how mortality differences between mating type alleles may alter the expected number of mating types in isogamous species. We examine a mathematical evolutionary model of isogamous populations of infinite size and find an analytic expression for the expected number of mating type alleles. We will derive general results that apply to models in which the mortality rates of mutant types are drawn from arbitrary probability distributions.

The paper is structured as follows. We begin in §2 by presenting the model that we will use for our analysis, which can be broken down into two components; a short time-scale model of the mating type population dynamics and longer time-scale model of the evolutionary dynamics. In §3.1, we analyse the behaviour of the short-term population dynamics, deriving conditions for the stability of the population. Section 3.2 uses these results to make predictions about the long-term evolutionary dynamics when mutation and extinction events occur. First examining a special case when mutant mortality rates are drawn from a delta distribution, we infer the distribution-dependent scaling factor of the expected number of mating type alleles using an integral transform. Finally, we conclude by exploring the biological implications of our results with reference to available empirical data.

# 2. Population and evolutionary models

In constructing a mathematical model for the evolution of mating types, we must determine appropriate descriptions of facultative sexual reproduction in a population, and of the emergence of mutations. Let us briefly discuss our choices, and their biological implications, with reference to the literature.

Some authors have modelled facultative sex as a population switching between purely sexual and asexual reproductive modes [28,34], while others consider a population average rate of asexual and sexual reproduction [20,22], with individuals able to engage in either mode at any given time. We take the latter approach as it does not require a population-level coordination in the selection of reproductive mode. The actual rate of sexual reproduction in a facultatively sexual population is probably a complex function of the ecology and population genetics of the species under consideration. Moreover, the exact theoretical mechanisms that generate selection for sexual reproduction are an intensely debated subject [35]. To understand the effect of the rate of sexual reproduction on the number of mating types, we treat it as a control parameter in our model.

Regarding mutation, we must distinguish between two relevant types of mutations that our model should account for: those that affect the (frequency-independent) fitness of individuals, and those that affect their mating behaviour. In principle, mutations affecting frequency-independent fitness may or may not be linked to the underlying mating type; however, in our model, only one of these is relevant. Unlinked mutations would become disassociated from the mating type background on which they arise via genetic recombination during sexual reproduction. Thus the effect of such unlinked mutations would be most clearly seen during periods of purely asexual reproduction in the population. Such dynamics would, therefore, be obscured in our chosen modelling framework, in which the timing of sexual reproduction is not correlated across the population. We, therefore, only consider mutations affecting frequency-independent fitness that are explicitly linked to the mating type locus.

Finally, we note that mutations that generate new mating type properties may be realized in different ways. Some studies consider the gradual emergence of new mating types from their self-incompatible ancestors, with intermediate stages accounted for in which the mutant mating type may not be fully compatible with all resident mating types [17]. However, the vast majority of studies take a simplified modelling approach, in which new self-incompatible mating type alleles are introduced to the population in a state fully compatible with all resident types. We will follow this second approach, but note that frequency-independent fitness mutations that are linked to mating type alleles could be interpreted in a loose sense as accounting for increased or decreased maladaption between a given mating type and the remaining residents.

We now proceed to describe the precise mathematical details of the model. The model is split into two distinct time scales: Short-term population dynamics and the long-term evolution of the mating types. We describe the short-term dynamics in terms of deterministic ordinary differential equations for the frequencies of the mating types in the population. These are assumed to reach a stable stationary state on a much faster time scale than mutations occur. By then introducing random mutation events, we explain how the system evolves in the long term.

## 2.1. Short-term population dynamics

The model for our short-term population dynamics is similar to that introduced in [20]. An allele at a single locus determines the mating type of a haploid individual (see appendix A for a full biological motivation). In order to investigate the evolution of mating type number, we allow an infinite number of alleles at that locus, i.e. there is *a priori* no upper bound for the number of different mating types in a population. The population dynamics are governed by the following types of events: asexual reproduction, sexual reproduction and death. The processes are depicted in figure 1. We use a Moran type modelling framework (the population size is fixed by coupled birth–death events and generations are non-overlapping), but take the limit of infinite population size.

When reproducing asexually, the individuals produce exact copies of themselves. We take a coarse-grained approach and consider a 'population average' rate of asexual and sexual reproduction (i.e. sexual reproduction is not correlated in time across the entire population). The rate at which a given mating type reproduces asexually is given by a fixed clonal reproduction rate $c$. For a population that only reproduces asexually, $c = 1$. By contrast, we have $c = 0$ for a population that only reproduces sexually, and values in between for facultative sex.

During a sexual reproduction event, which occurs at rate $(1 - c)$, individuals of two different mating types are required. Hence, the frequency of sexual reproduction depends on the frequencies of both mating types. We assume mass-action encounter rate dynamics. For simplicity, we further assume that each sexual pairing produces just one progeny. This offspring inherits the mating type of either parent with probability $1/2$.

As addressed in the introduction to this section, we assume that mutations that affect the frequency-independent fitness of a mating type are linked to the mating type allele. We now make a further simplification and assume that this fitness is predetermined and invariant with time. Differences in fitnesses between mating types are realized through a mating type specific mortality rate, $D_i$ (which can be interpreted as an inverse fitness of the $i$th type). Death events for a given type $i$ (which we recall are coupled to reproduction events) are then weighted by the rate $D_i$.

We now formulate the above model mathematically (for more details, we refer to the electronic supplementary material in [20]). Given a population with $M$ distinct mating types, we write $x = (x_1, \ldots, x_M)$ for the relative frequency of individuals with mating type $i$, so that $\sum_{i=1}^{M} x_i = 1$. The rate at which

**Figure 1.** Schematic reaction equations for the short-term population dynamics. Individuals reproduce asexually at rate $c$. Another individual is picked randomly according to the mortality rate $D_i$ linked to its mating type allele, and replaced by the offspring. The new individual inherits the mating type of the parent. Two individuals of different mating types mate at rate $1-c$. The offspring inherits the mating type of either parent with probability 1/2. Again, another individual is replaced in the population according to the mortality rates.

type $i$ replaces type $j$ is given by the matrix elements

$$T_{ij} = \left[ cx_i + (1-c)\frac{1}{2}x_i \sum_{k \neq i} x_k \right] D_j x_j. \tag{2.1}$$

The first term in the bracket describes the asexual reproduction with the clonal reproduction rate $c$. The second term describes the sexual reproduction between an individual of mating type $i$ and any other type which is not the same. Individuals mate with each other at rate $1-c$ and the offspring inherits the mating type $i$ with probability 1/2. Keeping the total population size fixed, reproduction is coupled with the removal of individuals from the population, with death rate $D_j$ for type $j$.

The population dynamics for this model are then described by the ODE system

$$\frac{dx_i}{dt} = \sum_j (T_{ij} - T_{ji})$$
$$= \frac{1-c}{2} \sum_j x_i x_j \left[ D_i x_j - D_j x_i + \frac{1}{\gamma}(D_j - D_i) \right], \tag{2.2}$$

where we have introduced the parameter

$$\gamma = \frac{(1-c)}{(1+c)}, \tag{2.3}$$

for mathematical convenience in the forthcoming analysis. Note that since we are working in the limit of infinite population size, the dynamics are entirely deterministic. Thus we ignore the possibility of rare types being lost through genetic drift (for studies analysing such stochastic invasion dynamics, see [21,22]). The only way that extinctions can occur in this model are when the dynamics of equation (2.2) drive one or more mating type frequencies to zero at long times.

The mortality rates $D_i$ are arbitrary positive parameters. We are interested in the effect of mortality differences between mating type alleles, not the evolution of the actual values themselves. Hence, we impose the rule $\sum_i D_i / M = 1$ and decompose

$$D_i = 1 - r_i, \tag{2.4}$$

where the $r_i$ are the *residuals* describing the difference between the average mortality rate of the population and that of type $i$. We interpret these values as representing *fitness*, since larger $r_i$ implies lower mortality rates and therefore higher reproductive success.

## 2.2. Long-term evolutionary dynamics

For the long-term evolution of the system, we consider mutations and extinctions of mating types. When a mutation event occurs, a novel mating type is introduced to the population. The novel mating type obeys the same dynamical rules as the resident population, but has a distinct mortality rate. Note that

by assuming this mortality rate is coupled to the mating type allele and invariant in time, we are able to characterize the model in terms of a single mutation rate, the arrival rate of new mating type alleles. The precise value for the mutant mortality rate is drawn from an arbitrary distribution $p$. From a theoretical standpoint, the choice of such a distribution poses a common problem, with empirical data suggesting that fitness distributions vary across species [36]. We, therefore, make no *a priori* assumptions about the form of this distribution and instead proceed with the aim of deriving results for arbitrary distributions.

We next assume a low mutation rate, so that the short-term population dynamics is always in a stationary state upon the introduction of a new mutant mating type. From a consideration of the form of equation (2.2), it is clear that the fixed points of the short-term population dynamics (along with their stability) will be dependent on the precise values of the mortality rates. We will characterize these properties mathematically in §3.1. Here, we note that upon introduction of a mutant mating type there are only a limited class of events that can occur.

Suppose a novel mating type $\mu$, with frequency $x_\mu$, is introduced to the population. If the short-term dynamics approach, a stable interior (i.e. $0 < x_i < 1$ for $i = 1, 2, \ldots, M, \mu$) fixed point, we interpret the mutant as having successfully invaded, such that it is then part of the resident population:

$$x \mapsto (x_1, \ldots, x_M, x_\mu), \quad M \mapsto M + 1. \tag{2.5}$$

If, however, the fixed point is not interior (i.e. $0 < x_i < 1$ for $i = 1, 2, \ldots, M, \mu$), the short-term dynamics must tend to a fixed point in the system, where at least one of the mating type frequencies will be reduced to zero population density (i.e. at least one of the mating types will go extinct). We first consider the situation where just one of the mating types, $e$, is extinct. Note that under the symmetry of the mating dynamics, the mating type with the lowest frequency (i.e. $x_e = 0$) will be that with the highest mortality. We use this fact to identify the extinct mating type $e$, and remove it from the population:

$$x \mapsto (x_1, \ldots, x_{e-1}, x_{e+1} \ldots), \quad M \mapsto M. \tag{2.6}$$

We note that at this stage the number of mating types has not changed, with the mutant mating type either having replaced a resident or been itself driven to extinction. A new fixed point exists. If the new stable fixed point is interior (i.e. $0 < x_i < 1$ for $i \neq e$), we can proceed to introduce another mutation and repeat the above process. If the new fixed point is not interior (i.e. $0 < x_i < 1$ for $i \neq e$), we must seek the next mating type $e'$ to go extinct. By the same logic as above, this will be the mating type with the highest mortality among the remaining mating types:

$$(x_1, \ldots, x_{e-1}, x_{e+1} \ldots) \mapsto (x_1, \ldots, x_{e-1}, x_{e+1} \ldots, x_{e'-1}, x_{e'+1} \ldots), \quad M \mapsto M - 1. \tag{2.7}$$

This process is repeated until the short-term dynamics succeeds in reaching a stable fixed point where coexistence is possible. The general algorithm is illustrated in figure 2.

As mentioned before, we are not interested in the evolution of the mortality rates $D_i$ themselves. Therefore, after each mutation and extinction event, we re-centre the mortality rates around a mean value of one. For this, we use the decomposition into fitnesses $r_i$ in equation (2.4). Let $r_1, \ldots, r_M$ be the fitnesses for the residing mating types in the population. Note that the fitnesses, $r_i$, sum to zero by definition. After adding a mutant with value $r_{M+1}$, we obtain the new fitnesses

$$r_i' = r_i - \frac{1}{M+1} \left[ \underbrace{\sum_{j=1}^{M} r_i}_{=0} + r_{M+1} \right] = r_i - \frac{r_{M+1}}{M+1}. \tag{2.8}$$

Similarly, after removing an extinct mating type with the lowest fitness $r_{\min} = \min(r_i)$ from the population, we obtain

$$r_i' = r_i - \frac{1}{M-1} \left[ \underbrace{\sum_{j=1}^{M} r_i}_{=0} - r_{\min} \right] = r_i + \frac{r_{\min}}{M-1}. \tag{2.9}$$

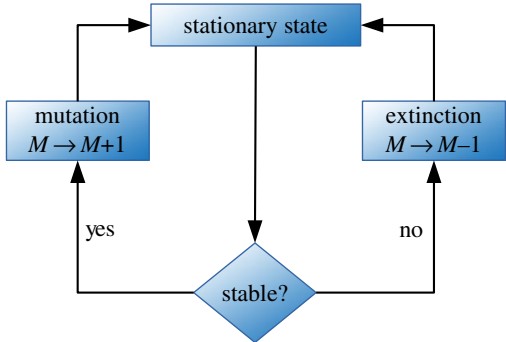

**Figure 2.** Schematic algorithm for the long-term population dynamics. First, we compute the stationary state of the short-term system according to equation (3.1). Then, we check the stability of the fixed point using the criteria in equation (3.2). If the fixed point is stable (i.e. mating types coexist), a mutation event occurs and we add a novel mating type to the population. If the fixed point is not stable, the mating type with the highest mortality rate goes extinct, i.e. is removed from the population. After each mutation and extinction event, we compute the new stationary state.

## 3. Results

In the long-term evolutionary model, the number of mating types $M$ is a random variable which evolves over time. Simulations show that we can expect the process to reach a stationary distribution with the number of mating types fluctuating close to some mean value. Our goal is to calculate this value. In particular, we aim to predict the average number of mating types in terms of the model parameters, including the fitness variability. We now proceed with the full mathematical analysis. For readers less concerned with the specific mathematical details of the derivation, our key result is given in equation (3.21).

In order to determine how the system evolves in the long term, we first need to know how the short-term population dynamics behave. For this, we analyse the stationary states of the short-term dynamical system and prove conditions for stability in terms of the fitnesses $r_i$ (see theorem 3.1). In analysing the long-term dynamics, we seek to link the spread in fitness (described by the distribution $p$ from which the mortality rates of new mutants are drawn) to the expected number of mating types at long times, $\mathbb{E}_p M$. Our analysis will proceed by first studying the special case that new mutants always have the same fitness advantage over the residents (i.e. $p$ is a Dirac delta distribution), then applying a heuristic to expand these results to the general case.

### 3.1. Short-term dynamics: stable fixed points

The dynamics of the short-term model with no variation in fitness are simple: when $D_i \equiv 1$, the mating type frequencies approach the stable fixed point $x_i^\star = 1/M$. In the general heterogeneous case, however, simulations show that the fixed point destabilizes with increasing mortality rate variability (figure 3). For the coexistence of mating types in a population, we require the short-term dynamics to reach a stable stationary state with non-zero mating type frequencies. Otherwise, the frequencies of the mating types with lower fitness values would eventually reach zero, i.e. the mating types go extinct. Hence, we ask if a stable fixed point exists and under which conditions with respect to the range of allowed fitness values.

**Theorem 3.1**

(i) *The ODE system in equation (2.2) has a fixed point $x^\star$ given by*

$$x_i^\star = \frac{1}{M}\left[1 + \left(\frac{M}{\gamma} - 1\right)r_i\right],$$
(3.1)

(ii) $x^\star$ *is stable if it is interior, that is, if $x_i \in (0, 1)$ for all $i$, and*
(iii) $x^\star$ *is interior if and only if the fitnesses obey the inequalities*

$$-R(M) < r_i < (M-1)R(M),$$
(3.2)

*where we have introduced $R(M) = (M/\gamma - 1)^{-1}$.*

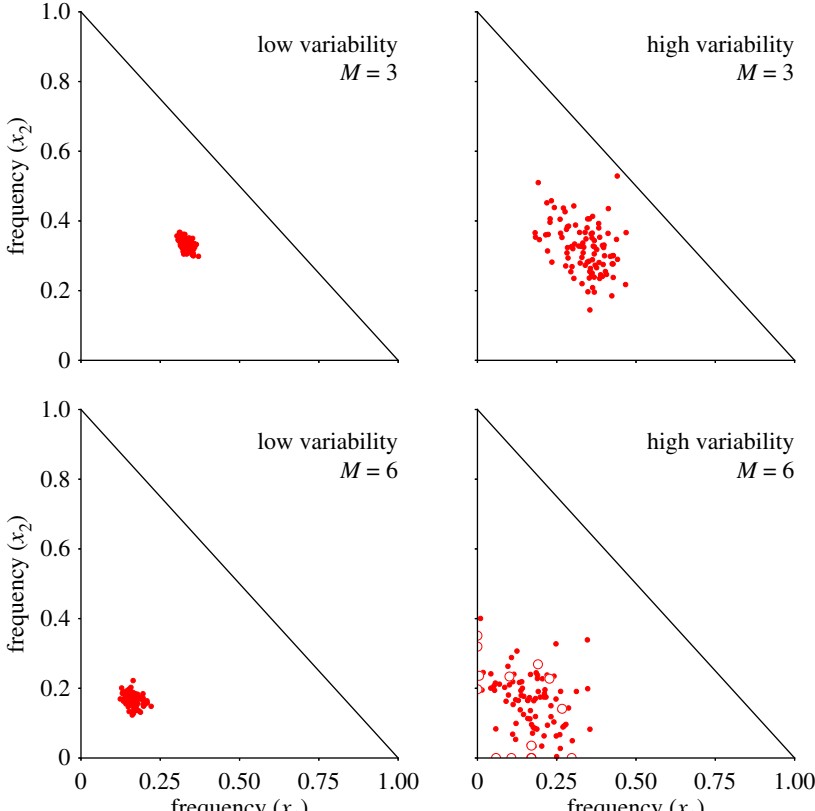

**Figure 3.** Fixed points computed from samples of mortality rates. Due to the assumption of fixed population size, for a population with $M = 3$ mating types there are only two free variables, the frequency of mating types 1 and 2 ($x_1$ and $x_2$, respectively). Similarly, for a population with $M = 6$ mating types there are five free variables. For clarity, we plot the position of the fixed points $\boldsymbol{x}^* = (x_1, \ldots, x_5)$ as a projection in ($x_1$, $x_2$) plane. Mortality rates are drawn from a normal distribution with low variance 0.001 (left) and high variance 0.005 (right) and the clonal rate is set to $c = 0.9$. The fixed point destabilizes as the fitness variability increases (left to right). Frequencies are pushed towards the extinction boundary as the number of mating types $M$ increases (top to bottom). Fixed points that lie on an extinction boundary ($x_i = 0$) in the higher-dimensional space (i.e. where one or more mating types is extinct) are indicated as empty circles.

*Proof.* We first compute the non-trivial fixed point. Noting that since the addition of a constant factor will not change our conclusions about the fixed point or its stability, we rescale time using $t \mapsto (2/(1-c))t$ and write

$$\frac{\mathrm{d}x_i}{\mathrm{d}t} = f_i(\boldsymbol{x}) := \sum_j x_i x_j g_{ij} \tag{3.3}$$

with

$$g_{ij}(\boldsymbol{x}) = D_i x_j - D_j x_i + \frac{1}{\gamma}(D_j - D_i). \tag{3.4}$$

Since $\sum_i r_i = 0$, it is easy to check that the fixed point $\boldsymbol{x}^\star$ proposed in (3.1) obeys $\sum_i x_i^\star = 1$ as required. From (3.3), it will, therefore, suffice to show that $g_{ij}(\boldsymbol{x}^\star) = 0$ for all $i$, $j$. This is an over-specified linear system, and it is straightforward to check that

$$D_i x_j^\star - D_j x_i^\star = \frac{D_i}{M}\left[1 - \left(\frac{M}{\gamma} - 1\right)(D_j - 1)\right] - \frac{D_j}{M}\left[1 - \left(\frac{M}{\gamma} - 1\right)(D_i - 1)\right]$$

$$= \frac{1}{\gamma}(D_i - D_j),$$

and hence the result of equation (3.1) follows.

Next, we undertake a linear stability analysis of equation (3.3) around $x^\star$. The Jacobian matrix has entries

$$J_{ij} = \left.\frac{\partial f_i}{\partial x_j}\right|_{x^\star} = \sum_k \left( \delta_{ij} x_k^\star g_{ik}(x^\star) + \delta_{kj} x_i^\star g_{ik}(x^\star) + x_i^\star x_k^\star \left.\frac{\partial g_{ik}}{\partial x_j}\right|_{x^\star} \right). \tag{3.5}$$

Now,

$$\frac{\partial g_{ik}}{\partial x_j} = \delta_{kj} D_i - \delta_{ij} D_k, \tag{3.6}$$

and by construction $g_{ik}(x^\star) = 0$, so in fact we have

$$J_{ij} = \begin{cases} \left( D_i x_i^\star - \sum_k D_k x_k^\star \right) x_i^\star & \text{if } i = j \\ D_i x_i^\star x_j^\star & \text{if } i \neq j. \end{cases}$$

Note that, since $\sum_i x_i = 1$, we know $J$ must be a degenerate matrix having a zero eigenvalue. To show linear stability of $x^\star$, we are required to prove that the other eigenvalues of $J$ all have negative real part. It is well known (e.g. [37]) that all eigenvalues of a matrix lie inside the union of the *Gershgorin discs* in the complex plane; disc $i$ has centre given by the $i$th diagonal element of the matrix and a radius equal to the sum of the absolute values of the other entries of row/column $i$. The radii of the Gershgorin discs for our Jacobian matrix can be computed as follows:

$$\rho_i = \sum_{j \neq i} |J_{ji}| = \sum_{j \neq i} D_j x_j^\star x_i^\star = x_i^\star \left( \sum_j D_j x_j^\star - D_i x_i^\star \right). \tag{3.7}$$

With the diagonal element as the centre of disc $i$, we find that if $\lambda$ is an eigenvalue of $J$, we must have $\text{Re}[\lambda] \leq J_{ii} + \rho_i = 0$. We know of the existence of one zero eigenvalue of $J$; to prove stability we must establish its uniqueness.

If we assume that $x^\star$ is interior, the Jacobian matrix $J$ is *Metzler*, meaning that it has strictly positive off-diagonal entries. Addition of a sufficiently large constant to the diagonal of $J$ would give it all positive entries. The Perron–Frobenius theorem states that a real matrix with all strictly positive entries has a unique, real, rightmost eigenvalue. Therefore, $J$ also has a unique rightmost eigenvalue, which in this case must be zero, and hence $x^\star$ is linearly stable.

Let us now deduce the criterion for the fixed point $x^\star$ to be interior, in terms of the difference in mortality rates between mating types. At one end of the allowed range, we obtain

$$\frac{1}{M} \left[ 1 + \left( \frac{M}{\gamma} - 1 \right) r_i \right] = x_i^\star > 0,$$

$$\Rightarrow r_i > \frac{-1}{M/\gamma - 1} = -R(M).$$

and at the other

$$\frac{1}{M} \left[ 1 + \left( \frac{M}{\gamma} - 1 \right) r_i \right] = x_i^\star < 1,$$

$$\Rightarrow r_i < \frac{M - 1}{M/\gamma - 1} = (M - 1)R(M),$$

as required. ∎

## 3.2. Analysis of long-term model

Let us first discuss the general behaviour of the long-term evolutionary dynamics of our model (described in §2.2). In each simulation step, we introduce a new mutant and compute the fixed point of the initial system and check its stability according to the conditions given in equation (3.2). There are three cases to discuss:

**Failure.** If the mortality rate of the mutant mating type allele is too high relative to the existing population, no stable interior fixed point exists. In other words, the mutant does not invade the population, and the number of mating types does not change ($M \mapsto M$).

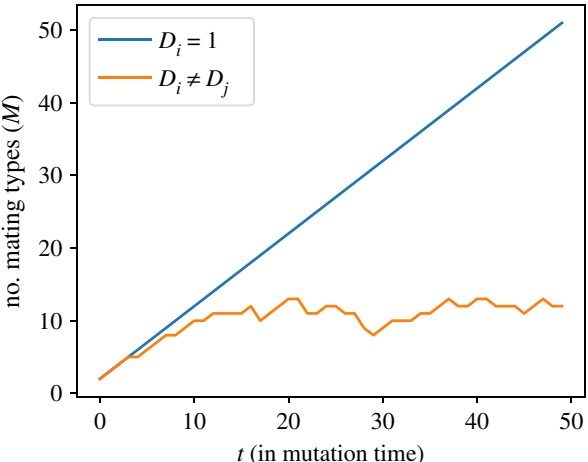

**Figure 4.** Time series of the number of mating types for clonal rate $c = 0.82$. In a population without mortality differences ($D_i = 1$), the number of mating type alleles grows consistently after each mutation event. With mortality rate differences ($D_i \neq D_j$), extinction events become more likely. Thus, we see a limited growth of the number of mating type alleles. The mutant mortality rates are drawn from a normal distribution with variance 0.001.

**Invasion with coexistence.** If the mortality rate of the mutant is not too high or too low, an interior fixed point may exist. In that case, the mutant mating type coexists with the residing types in the population. Thus, the number of mating type alleles increases by one ($M \mapsto M + 1$).

**Invasion with extinction.** For very fit mutants (i.e. low mortality rate), it is possible that no stable fixed point exists with all $M + 1$ types. Hence, one or more of the residing mating type alleles with the highest mortality rates go extinct. Potentially, a mutant can wipe out the whole population. The number of mating type alleles is reduced by the number of extinct residents ($M \mapsto M + 1 - M_{\text{extinct}}$). Note that in the case of a single extinction event, the resident mating type is simply replaced by the mutant, hence the number of mating type alleles does not change.

After each mutation event, we record the number of mating types residing in the population. This way, we obtain a time series of the evolution of the number of mating types $M$ as in figure 4. In the homogeneous case, where all mortality rates are equal ($D_i = 1$), we see a constant growth of the number of mating types as expected. A stable interior fixed point is always reached and thus, mutant mating type alleles invade the population. With mortality differences ($D_i \neq D_j$), the growth of the number of mating types is limited. The fixed point destabilizes and mating types with higher mortality rates are pushed towards extinction. Higher asexual reproduction rates $c$ can amplify the extinction rate. Thus, we see fewer mating types as $c \to 1$. In the long term, the average number of mating types approaches a specific value, as we discuss in the following.

From simulations, we observe that the average number of mating types changes according to the model parameter $\gamma$, which is a measure of the rate of sex in the population (see equation (2.3)). Meanwhile, changing the distribution $p$ from which the fitnesses ($r$) of new mutants are drawn appears to affect the average number of mating types only through scaling by a constant factor (see figure 5). This relation is demonstrated in figure 5, where we show the average number of mating types obtained from arbitrary fitness distributions $p$ as examples. This observation motivates us to seek a functional $\phi[p]$ depending on the fitness distribution $p$ such that $\phi[p] \cdot \mathbb{E}_p M$ is invariant. That is, for some function $m(\gamma)$ we have

$$\phi[p] \cdot \mathbb{E}_p M = m(\gamma). \tag{3.8}$$

To find the scaling functional, we use a heuristic approach based on two assumptions: (i) only beneficial mutations ($r > 0$) are relevant, since deleterious mutations are very unlikely to invade, and (ii) the functional $\phi$ can be obtained as the average of some function of fitness. The consequence of these assumptions mathematically is that we may write the integral transform

$$\phi[p] = \frac{\int_0^\infty k(r) p(r) \, dr}{\int_0^\infty p(r) \, dr} . \tag{3.9}$$

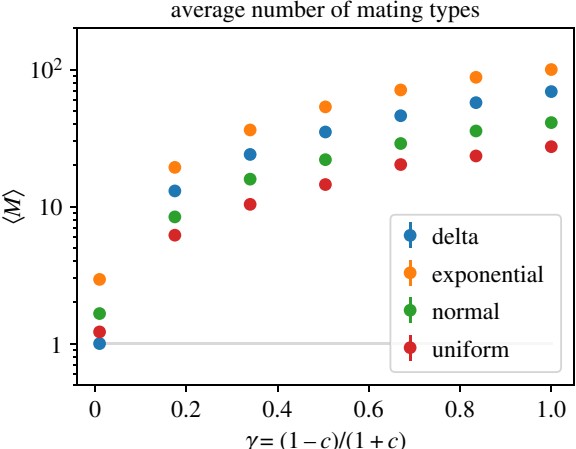

**Figure 5.** Average number of mating types against the model parameter $\gamma = (1-c)/(1+c)$ for mutant mortality rates drawn from different distributions; (i) Dirac-delta distribution with peak at $\bar{r} = 0.02$, (ii) exponential distribution with scale parameter $\beta = 0.01$ and left-shift $r_b = -0.01$, (iii) normal distribution with mean $\mathbb{E}[r] = 0.001$ and variance $\mathrm{Var}[r] = 0.001$, (iv) uniform distribution with mean $\mathbb{E}[r] = 0$ and variance $\mathrm{Var}[r] = 0.002$.

for some kernel function $k$. To compute the scaling functional, we need to determine the kernel function $k(r)$. Since $k$ does not depend on $p$, we are free to choose a solvable model to analyse.

### 3.2.1. Non-random mutant mortality rates

In the previous section, we found that the missing quantity we needed to determine the number of mating types predicted by our model was a kernel function $k(r)$. However, we also saw that we were free to determine this function using any distribution of mating type fitnesses, $p$. In this section, we, therefore, choose $p$ to have a particularly simple form, that of the delta-distribution. Let $p = \delta_{\bar{r}}$ be a Dirac distribution centred at position $\bar{r} > 0$. Note that this implies a degenerate case where fitnesses are non-random; the Dirac distribution always returns mutants with a fitness of precisely $\bar{r}$ (i.e. the delta-distributed fitness values have mean fitness $\bar{r}$ and variance zero). We emphasize that this distribution is not biologically relevant, but will provide a route to obtaining general analytical results for more realistic distributions.

According to equation (3.9), we have

$$k(\bar{r}) = \phi[\delta_{\bar{r}}]. \tag{3.10}$$

If we can compute an expression for the expected number of mating types of the functional form

$$\mathbb{E}_{\delta_{\bar{r}}} M = \frac{m(\gamma)}{k(\bar{r})}, \tag{3.11}$$

then comparing with equation (3.10) should allow us to determine the scaling functional for generic distributions, which can be constructed as the superposition of Dirac delta distributions.

*Stationary configuration of fitnesses.* For mutant mortality rates drawn from a delta distribution, simulations imply that the system is driven towards a stationary configuration of fitnesses. Figure 6 shows an example time series of the fitnesses $r_i$ linked to the mating type alleles residing in the population. From simulations, we observe that there appears to exist an absorbing stationary configuration in which the fitnesses are equally spaced.

For this configuration to be stable under the addition of a new mutant requires that the invader replaces the residing mating type with the lowest fitness, such that (after re-centring) the same equispace configuration is recovered. It follows that we must, therefore, have $\bar{r} > \max(r_i)$.

To compute this state, we write the fitnesses as ranked values,

$$r_1 > r_2 > \cdots > r_{M-1} > r_M,$$

(i.e. $r_1$ is the highest fitness value in the population). Since the values are centred at 0, we have $r_M < 0 < r_1$. In addition, we assume that there are no duplicate fitness values, which is ensured when mortality rates are drawn from a delta distribution.

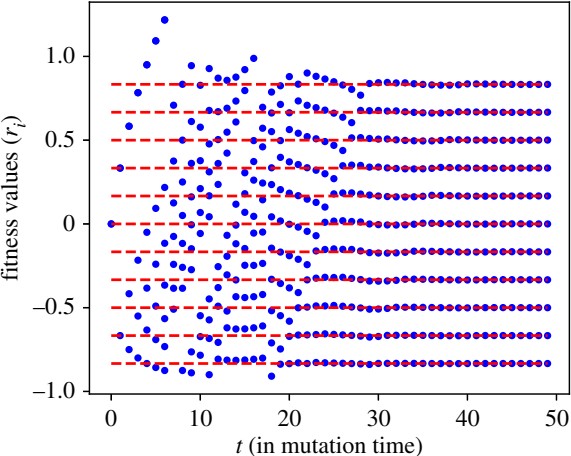

**Figure 6.** Time series of fitnesses for mutant mortality rates drawn from a delta distribution with peak at $\bar{r} = 0.001$. The clonal rate is set to $c = 0.82$. Fitnesses evolve towards a stationary configuration. Dashed lines indicate the analytically derived stationary configuration.

Now let us discuss the stability of the fitness configurations before and after a mutation event. Since the stationary configuration must be stable, we require according to equation (3.2) that $r_M > -R(M)$. After adding a mutant with $r_0 > r_1$, we compute the new fitnesses and obtain with equation (2.8)

$$r_i' = r_i - \frac{r_0}{M+1}, \tag{3.12}$$

with $i = 0, \ldots, M$. For a stationary configuration, the mating type with the largest mortality rate must go extinct, i.e. we require an unstable fixed point with $r_M' < -R(M+1)$. After eliminating the mating type with the fitness $r_M'$, we compute the new fitnesses again. Here, we use equation (2.9) and insert the result from equation (3.12) to obtain

$$r_i'' = r_i' + \frac{r_M'}{M}$$
$$= r_i - \frac{r_0}{M+1} + \underbrace{\frac{1}{M}\left(r_M - \frac{r_0}{M+1}\right)}_{\Delta r}, \tag{3.13}$$

with $i = 0, \ldots, M-1$.

Since we assume a stationary state, we have

$$r_i = r_{i-1}'' = r_{i-1} - \Delta r = r_0 - i\Delta r,$$

with $i = 1, \ldots, M$. Here, we see that the fitnesses are equally spaced with distance $\Delta r = (r_0 - r_M)/M$. To compute the distance, we use that the sum of the fitnesses is zero, i.e. we have

$$0 = Mr_0 - \sum_{i=1}^{M} i\Delta r = Mr_0 - \frac{M(M+1)}{2}\Delta r.$$

We solve for $\Delta r$ and obtain

$$\Delta r = r_0 \frac{2}{M+1}. \tag{3.14}$$

Finally, for the stationary configuration of fitnesses, we have

$$r_i = r_0\left(1 - \frac{2i}{M+1}\right). \tag{3.15}$$

The stationary configuration we derived analytically is also plotted in figure 6.

*Expected number of mating types.* To compute the expected number of mating types, we must check which values of $M$ admit stationary configurations of the form derived above. The conditions for a stationary state from theorem 3.1 deliver bounds on the allowed values of $M$. Let $r_0 = \bar{r}$ be the mutant

mortality rate drawn from a delta distribution $p \sim \delta_{\bar{r}}$. Using equation (3.15), we have

$$\bar{r}\frac{M-1}{M+1} < \frac{1}{M/\gamma - 1}.$$

We solve for $M$ and obtain the upper bound,

$$M < M^+ = \frac{1}{2}\left[\gamma\left(\frac{1}{\bar{r}}+1\right)+1+\sqrt{\left(\gamma\left(\frac{1}{\bar{r}}+1\right)+1\right)^2+4\gamma\left(\frac{1}{\bar{r}}-1\right)}\right]. \tag{3.16}$$

Likewise, the opposite limit delivers

$$\bar{r}\frac{M}{M+1} > \frac{1}{(M+1)/\gamma - 1},$$

and we obtain the lower bound

$$M > M^- = \frac{1}{2}\left[\gamma\left(\frac{1}{\bar{r}}+1\right)-1+\sqrt{\left(\gamma\left(\frac{1}{\bar{r}}+1\right)-1\right)^2+4\frac{\gamma}{\bar{r}}}\right]. \tag{3.17}$$

We observe that, for different values of $\bar{r}$ and $\gamma$, the region of possible numbers of mating types implied by the above bounds may contain more than one integer. However, for small $\bar{r}$ (corresponding to small fitness differences between mating types) we have the first-order scaling law

$$M^{\pm} = \frac{\gamma}{\bar{r}} + \mathcal{O}(1). \tag{3.18}$$

Hence we identify

$$\mathbb{E}_{\delta_{\bar{r}}}[M] \sim \frac{\gamma}{\bar{r}}. \tag{3.19}$$

Figure 7 shows a contour plot of the expected number of mating types (approximated by $(M^+ + M^-)/2$) for different combinations of the model parameter $\gamma$ and the delta distribution peak $p = \delta_{\bar{r}}$. For low sex rates ($c \to 1$), the number of mating type alleles is small. As the mortality differences $\Delta r \sim \bar{r}$ increase, the number of mating types is lowered further.

In figure 8, we compare the analytically derived expected number of mating types in equation (3.19) with simulations. We see that the simulated number of mating types converges to the expected value. Hence, our assumptions about the stationary configuration of the fitnesses and the conclusion we draw for the number of mating types are validated.

### 3.2.2. Average number of mating types for general mortality distributions

We now compare the result of the previous section, equation (3.19) with equation (3.11) to determine the simple rules $m(\gamma) = \gamma$ and $k(r) = r$. For general fitness distributions, we, therefore, expect the scaling function of equation (3.9) to be given by

$$\phi[p] = \frac{\int_0^{\infty} r p(r)\,dr}{\int_0^{\infty} p(r)\,dr} = \mathbb{E}_p[r|r > 0]. \tag{3.20}$$

In words, the scaling functional determining the expected number of mating types is simply the mean fitness value of beneficial mutations. Finally, we obtain an approximation for the average number of mating types:

$$\mathbb{E}_p M \approx \left(\frac{1-c}{1+c}\right)\frac{1}{\mathbb{E}_p[r|r > 0]}. \tag{3.21}$$

Let us compare our analytic prediction of the average number of mating type alleles with values obtained from simulations. Figure 9 shows the average number of mating types as a function of the clonal rate $c$ for different mortality rate distributions. We see that the analytic result in equation (3.21) predicts the average number of mating types very well, especially for small values of $c$.

Noting that we have defined the parameter $\gamma = (1-c)/(1+c)$, as $c \to 1$ we would expect our approximation to break down as higher order terms in the scaling law equation (3.18) become non-negligible. Indeed, intuitively we note that while the number of mating type alleles is discrete and

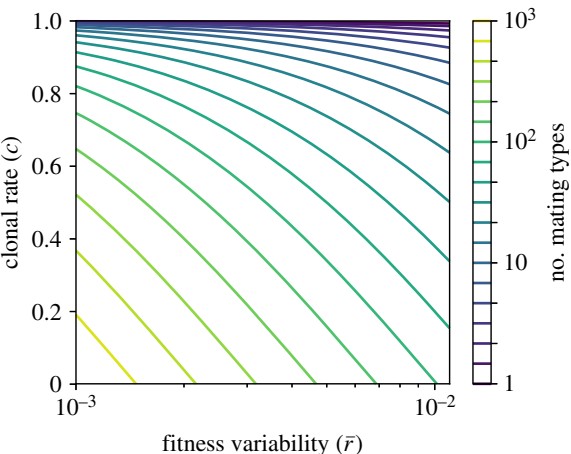

**Figure 7.** Contour plot of the expected number of mating type alleles $\mathbb{E}_p M$ in equation (3.19) for mutant mortality rates drawn from a delta distribution. The expected number of mating types depends approximately on the ratio of the model parameter $c$ and the peak position $\bar{r}$ of the delta distribution.

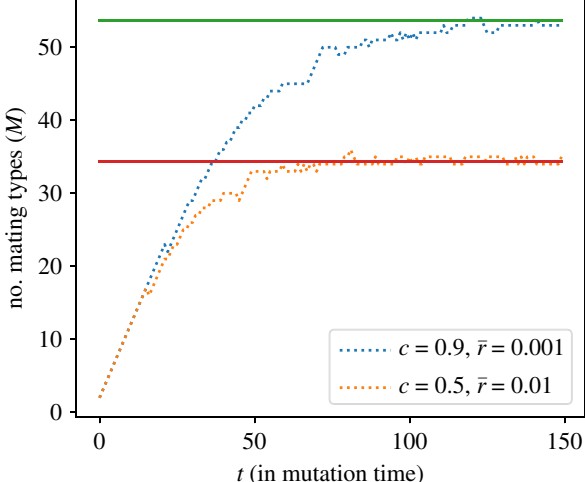

**Figure 8.** Time series of the number of mating type alleles $M$ for mutant mortality rates drawn from delta distributions with different peak positions $\bar{r}$ (dashed lines). Solid lines indicate the analytically derived expected number of mating types in equation (3.19).

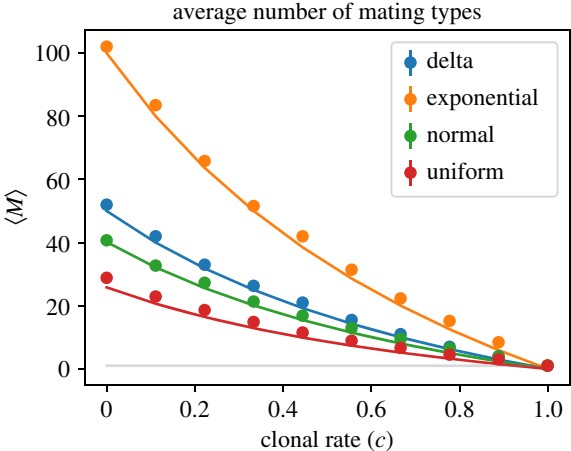

**Figure 9.** Average number of mating types for mutant mortality rates drawn from different distributions (same parameters as in figure 5) against the model parameter $c$. Solid lines correspond to the analytically derived prediction in equation (3.21). Note that the minimum number of mating type alleles is 1 (grey line); our prediction does not respect this limit as $c \to 1$, although this is not visible on the scale of the figure.

bounded below by one, equation (3.21) is continuous and takes values in $[0, \infty)$. Nevertheless, as demonstrated in figure 9, the approximation continues to provide good estimates for the expected number of mating types.

# 4. Discussion

In this paper, we have addressed the problem of why the number of mating types in isogamous species is often low, when naive evolutionary reasoning suggests there should be very many types, each at very low frequency. To this end, we analysed a simple model that incorporates differences in fitness between mating type alleles. Our main result, equation (3.21), shows that unless mating type mutants have precisely equal fitness to their ancestors, the number of mating types in a population will not grow indefinitely, but rather plateau at a finite value.

In a biological sense, one of the most interesting features of equation (3.21) is its dependence on the rate of asexual to sexual reproduction, $c$. As this parameter increases, the average number of mating types in the population decreases for any choice of mutant mating type fitness distribution. This is in line with previous observations that the rate of facultative sex may be a key empirical predictor of the number of mating types observed in isogamous species [20,22]. The number of mating types also decreases as the variance of the mutant mating type fitness from its ancestor increases. We can then see that when sex is very rare, mating type mutants must have extremely similar fitnesses for the maintenance of multiple mating types to be possible. For instance, if sex occurred of the order of once every thousand generations ($c = 0.999$), as has been recorded in some yeasts and algae [38,39], mutant fitness (measured by relative death rate) would need to have a variance of less than $10^{-6}$ from their ancestors if drawn from a normal distribution for even two mating types to be observed (figure 5).

We may then ask what empirical support exists for fitness differences between mating type alleles. As we have demonstrated in figure 3, we would expect prominent deviations from an even ratio ($x^*_i = 1/M$) in the rare sex regime. Thus skewed mating type ratios may provide evidence for variability in mating type fitnesses. Such skewed ratios have been recently observed in *D. discoideum* [40], as well as the isogamous species *Cryptococcus neoformans* [41] and *Candida glabrata* [42] (both with two mating types). Meanwhile indirect support for differences in mating type fitness comes from evidence of the repeated loss of mating type alleles in species phylogenies. Such losses appear to have occurred many times in ciliates [9] and fungi [43]. As it is possible that these extinctions could also be driven by genetic drift (demographic stochasticity) alone [20], we next turn to the potential causes of differential fitness between mating types.

Beneficial mutations on specific mating type alleles have been shown to be particularly prevalent in pathogenic fungi [43,44], where they can imbue increased virulence. Meanwhile, deleterious mutations have also been shown to accumulate more rapidly on mating type alleles as a result of suppressed genetic recombination at the mating type locus [31]. Within facultative sexuals, beneficial mutations at unlinked loci may also play a role. If a highly beneficial mutation arises during a period of asexual reproduction, the single mating type associated with this mutation may sweep to fixation before the next round of sexual reproduction. Indeed, this behaviour has been observed experimentally in *Chlamydomonas reinhardtii* [45]. Our model, which assumes a constant fitness for each novel mating type and an average rate of sexual reproduction across the population, takes a coarse-grained approach to these dynamics. Accounting for these additional processes (the accumulation of mutations on mating type alleles and switching between asexual and sexual environments) in a precise manner would be an interesting area for future investigation, but one that would require the specification of two mutation time scales (one for the arrival rate of new mating types and another for the mutations affecting the fitness of extant mating type alleles). Nevertheless, we would expect the qualitative picture to remain the same.

From a mathematical perspective, our techniques for modelling and analysing the effect of mutation differ from many standard approaches. Perhaps the most ubiquitous is adaptive dynamics [46], which considers the outcome of evolutionary dynamics in a continuous trait space [47] (that is, very small mutational steps are assumed [48]). However, such an approach is problematic when the quantity of interest (in our case, the average number of mating types) is explicitly a function of the size of the mutational steps. For instance, in equation (3.19), we find that if we take the limit of infinitely small mutational steps (i.e. we take the limit $\bar{r} \to 0$), the model predicts infinitely many mating types at leading order. Thus it is not clear that an adaptive dynamics approach would capture the interesting and biologically relevant dynamics revealed by our analysis.

In cases such as those described above, modelling discrete, randomly chosen mutations is more appropriate. However, one must then decide what distribution these should be sampled from, whether it be delta [20], normal [49], exponential [50] or generalized. As empirical fitness distributions have been shown to vary between species (and even between genetic loci) [36], the qualitative model dynamics must be independent of such a choice for any strong biological conclusion to be drawn. In this paper, we have shown that a heuristic approach, based on the assumption of the existence of a scaling factor, can provide a way to make analytic conclusions without specifying any precise form for the distribution of mutations. Interestingly, a similar observation has been made for a game theoretic model of the evolution of average population fitness [51], although the analytic form of the observed scaling function was distinct. An important avenue for further investigations will be to determine if this approach can be used as a general method for addressing problems involving stochastic mutation events.

In the shorter term, the approach we have developed will be useful for other problems in which balancing selection plays a role. In particular, this includes the study of alternate self-incompatibility systems. In the model we analyse in this paper, we assume that mating types are determined by one of an infinite set of potential alleles at one single locus. While this is a plausible model for many isogamous species, alleles at two distinct loci control mating type expression in many fungal species [12,13]. Future research would, therefore, involve extending the model presented here to account for this genetic feature.

As addressed in the introduction, self-incompatibility is also prevalent in many plant systems and, as a result of suppressed recombination, deleterious mutations can also accumulate at the self-incompatibility locus [52]. As these species are in general obligately sexual (i.e $c = 0$), our analysis suggests that such plants would be able to tolerate larger variance in the fitness of self-incompatibility alleles than facultatively sexual species. Once again, more specific modelling would need to be conducted to obtain a precise mathematical prediction for these diploid species, in which the genetic determination of self-incompatibility can sometimes be complex [53]. However, as effective population sizes in these species are often small (of order $10^3$ [54]) genetic drift is likely to play a more important role. It will, therefore, be interesting to investigate whether the techniques we have developed here for populations of infinite size can be generalized to capture the effect of finite population size.

Data accessibility. Data and relevant code for this research work are stored in GitHub: https://github.com/YvonneKrumbeck/Mating-Type-Model and have been archived within the Zenodo repository: https://doi.org/10.5281/zenodo.3631274.
Authors' contributions. Y.K. undertook the mathematical analysis, performed all simulations and drafted the manuscript. G.W.A.C. and T.R. conceived and supervised the study, participated in analysis and critically revised the manuscript.
Competing interests. We declare we have no competing interests.
Funding. T.R. and Y.K. are supported by the Royal Society. G.W.A.C. is supported by the Leverhulme Trust.
Acknowledgements. T.R. and Y.K. gratefully acknowledge the support of the Royal Society. G.W.A.C. thanks Leverhulme Trust for support through the Leverhulme Early Career Fellowship.

# Appendix A. Mating types versus sexes

In this appendix, we provide a biological justification for the mathematical model presented in §2. While the model is the same as that introduced in [20], it is important to understand its relevance for typical isogamous species. This is in particular because features of their reproductive behaviour differ significantly from those of the more familiar mammalian system.

For clarity, in figure 10, we compare and contrast the sexual cycle of mammals (figure 10a) with those of a 'typical' isogamous species, for which our model would be appropriate (figure 10b). The key differences are as follows:

1. Mammals are multicellular organisms that spend most of their life cycle in a diploid state (with two sets of chromosomes). By contrast, many isogamous species are unicellular, and typically exist in a haploid state (with one set of chromosomes).
2. For mammals, every reproductive event is a sexual one. By contrast, many isogamous species primarily produce asexually, only entering the sexual phase of their life cycle at the onset of environmental stressors (e.g. falling nutritional levels).
3. On reaching sexual maturity, mammals produce haploid gametes of different morphological types (sperm and eggs). This morphological type is determined by the genetic type of the diploid parent;

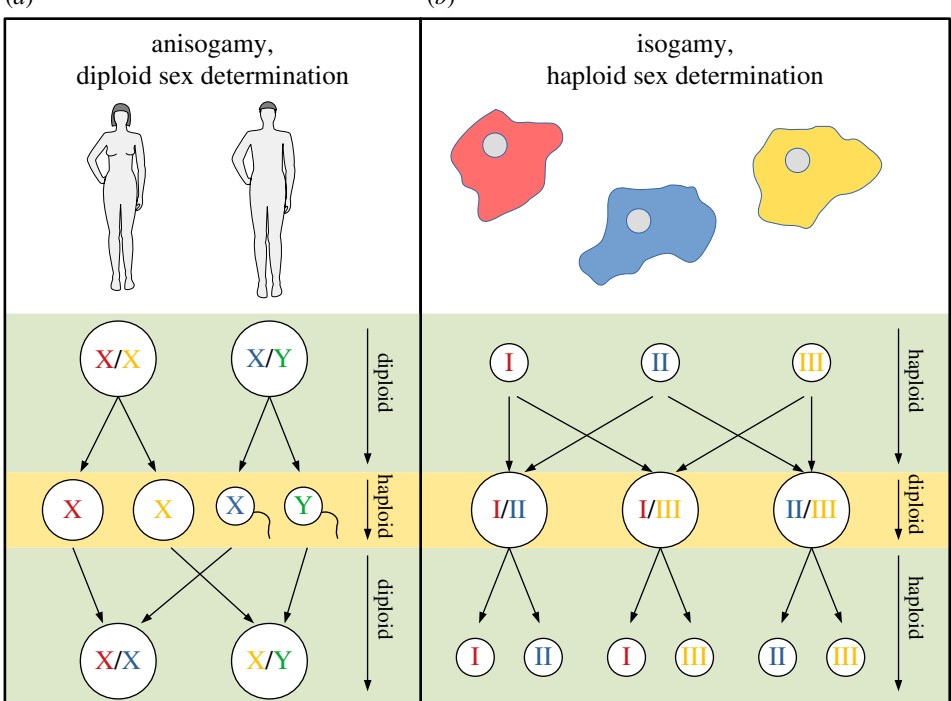

**Figure 10.** Comparison of the sexual life cycle of mammals when compared with a 'typical' isogamous species appropriate for our model. The life cycle of cells in mammals is dominated by a diploid state (green background). Here, the sex of the organism is determined by a pair of chromosomes (X/X in females and X/Y in males). The life cycle of cells in many isogamous species is dominated by a haploid state (green background). Here, an allele on a single chromosome determines the mating type (e.g. I, II and III in *Dictyostelium discoideum*). Mammals produce short-lived haploid sex cells (yellow background) that carry one chromosome type of the diploid parent. Their morphology differs according to the sex of their parent. A large egg cell and small sperm cell merge and form a diploid zygote which develops to the progeny. Cells of isogamous species, on the other hand, are equally sized and can reproduce asexually and sexually. During the sexual phase, two cells of different mating types merge to form a diploid zygote. From this short-lived diploid state (yellow background), the progeny emerge as haploid cells inheriting the parental mating types with equal probability.

females (with two X chromosomes) produce eggs carrying an X chromosome while males (with an X and a Y chromosome) produce sperm carrying an X or a Y chromosome. By contrast, for isogamous unicellular haploids, their entire cell transforms into a sexually capable gamete. The mating type of this haploid gamete is typically determined genetically by one of multiple alleles at a mating type locus. These gametes have similar morphology.

4. In mammals, the haploid sperm and egg fuse to form a diploid zygote. This zygote will grow to adult form and complete the life cycle. By contrast, the mating types of isogamous species can fuse to form a zygote in any non-self pairing (e.g. types I-II, I-III, II-III). This zygote cell is a transient state that will divide into a number of haploid daughter cells, with exactly half inheriting the mating type of either parent.

Our model takes a coarse-grained approach to some of these processes. For instance, while environmental stressors may be locally correlated in time for subpopulations of a given species, we have assumed an average rate of asexual to sexual reproduction, $c$, across a well-mixed population. We have also assumed that only one progeny is produced in every reproduction event, as opposed to multiple progeny. However, as addressed in §4, we expect the inclusion of such effects to lead to only qualitative differences.

The outline that we have presented above provides a picture of the life cycle features of a typical isogamous species. However, as with any general evolutionary model, there are examples of exceptions to this picture. For instance, while many isogamous species are unicellular (e.g. *S. cerevisiae*, *T. hyperangularis* and *C. reinhardtii*), some have a facultative multicellular life cycle (such as *D. discoideum*), while others are obligately multicellular (for example, *C. disseminatus* and *S. commune*). Furthermore, although haploidy is certainly the norm among isogamous algae and fungi, ciliates contain a diploid macronucleus (but exchange a haploid micronucleus during sexual

conjugation). Finally, while mating types are mostly inherited deterministically in a genetic manner, there are exceptions to this rule; many yeasts have evolved the ability to switch between mating types from generation to generation [55], while some ciliates inherit their mating type epigenetically or stochastically [9]. Models investigating the effect of these additional species-specific considerations are interesting, and indeed there have been a number of theoretical studies towards this end [28,34,56].

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
