## [Reviewer comments · Royal Society Open Science]

Review History

RSOS-192126.R0 (Original submission)

Review form: Reviewer 1

Is the manuscript scientifically sound in its present form?

Yes

Are the interpretations and conclusions justified by the results?

Yes

Is the language acceptable?

Yes

Do you have any ethical concerns with this paper?

No

Have you any concerns about statistical analyses in this paper?

No

Recommendation?

Accept as is

Comments to the Author(s)

.

Review form: Reviewer 2**Is the manuscript scientifically sound in its present form?**

Yes

Are the interpretations and conclusions justified by the results?

Yes

Is the language acceptable?

Yes

Do you have any ethical concerns with this paper?

No

Have you any concerns about statistical analyses in this paper?

No

Recommendation?

Accept with minor revision (please list in comments)

Comments to the Author(s)

The main result of the paper is demonstrating the role of fitness variation in determining the number of mating types emerging in simple evolutionary models. While it is a paper with motivation and interpretation on sexual reproduction, the analysis is based on mathematical models and assumptions. And that requires a very careful balance of a complete, overarching story and technical details (explaining why the reviewers had those issues), which the authors put great effort on before this resubmission.

In general, the authors have seriously responded to those comments and made the flow smoother throughout the manuscript. Unlike the two reviewers before with background on biology, I am a more mathematically minded reader and therefore the mathematical calculations, which are technically sound and beautiful, are within my capabilities to access. On the other hand, it took me much effort to digest the biological justification and arguments (they are significantly supplemented by the authors in this revised version). Therefore, I would suggest that the editor also refers to the comments from biologists again.

Additionally, the authors may want to double-check the text to eliminate any kind of grammar errors or typos. For instance, in Line 257 and Line 286, it may be more proper to use colons instead of semicolons.

Decision letter (RSOS-192126.R0)

08-Jan-2020

Dear Dr Rogers

On behalf of the Editors, I am pleased to inform you that your Manuscript RSOS-192126 entitled "Fitness differences suppress the number of mating types in evolving isogamous species" has been accepted for publication in Royal Society Open Science subject to minor revision in accordance with the referee suggestions. Please find the referees' comments at the end of this email.

The reviewers and handling editors have recommended publication, but also suggest some minor revisions to your manuscript. Therefore, I invite you to respond to the comments and revise your manuscript.

- Ethics statement

- Data accessibility

If you wish to submit your supporting data or code to Dryad (<http://datadryad.org/>), or modify your current submission to dryad, please use the following link:
<http://datadryad.org/submit?journalID=RSOS&manu=RSOS-192126>

- Competing interests

- Authors' contributions

AB carried out the molecular lab work, participated in data analysis, carried out sequence alignments, participated in the design of the study and drafted the manuscript; CD carried out the statistical analyses; EF collected field data; GH conceived of the study, designed the study,

coordinated the study and helped draft the manuscript. All authors gave final approval for publication.

- Acknowledgements

- Funding statement

Because the schedule for publication is very tight, it is a condition of publication that you submit the revised version of your manuscript before 17-Jan-2020. Please note that the revision deadline will expire at 00.00am on this date. If you do not think you will be able to meet this date please let me know immediately.

If your manuscript is newly submitted and subsequently accepted for publication, you will be asked to pay the article processing charge, unless you request a waiver and this is approved by Royal Society Publishing. You can find out more about the charges at <https://royalsocietypublishing.org/rsos/charges>. Should you have any queries, please contact openscience@royalsociety.org.

on behalf of Dr Kenta Ishimoto (Associate Editor) and Mark Chaplain (Subject Editor)
openscience@royalsociety.org

Associate Editor Comments to Author (Dr Kenta Ishimoto):

Associate Editor: 1

Comments to the Author:

I request of the authors that they should proofread the manuscript and correct typographical errors as suggested by one of the referees.

Reviewer comments to Author:

Reviewer: 1

Comments to the Author(s)

.

Reviewer: 2

Comments to the Author(s)

The main result of the paper is demonstrating the role of fitness variation in determining the number of mating types emerging in simple evolutionary models. While it is a paper with motivation and interpretation on sexual reproduction, the analysis is based on mathematical models and assumptions. And that requires a very careful balance of a complete, overarching story and technical details (explaining why the reviewers had those issues), which the authors put great effort on before this resubmission.

In general, the authors have seriously responded to those comments and made the flow smoother throughout the manuscript. Unlike the two reviewers before with background on biology, I am a more mathematically minded reader and therefore the mathematical calculations, which are technically sound and beautiful, are within my capabilities to access. On the other hand, it took me much effort to digest the biological justification and arguments (they are significantly supplemented by the authors in this revised version). Therefore, I would suggest that the editor also refers to the comments from biologists again.

Additionally, the authors may want to double-check the text to eliminate any kind of grammar errors or typos. For instance, in Line 257 and Line 286, it may be more proper to use colons instead of semicolons.

Author's Response to Decision Letter for (RSOS-192126.R0)

See Appendix A.

Decision letter (RSOS-192126.R1)

31-Jan-2020

Dear Dr Rogers,

It is a pleasure to accept your manuscript entitled "Fitness differences suppress the number of mating types in evolving isogamous species" in its current form for publication in Royal Society Open Science.

Kind regards,
Lianne Parkhouse
Editorial Coordinator
Royal Society Open Science

on behalf of Dr Kenta Ishimoto (Associate Editor) and Mark Chaplain (Subject Editor)
openscience@royalsociety.org

Appendix A

Dear Editor

We are very happy our manuscript has been accepted for publication in Open Science. On the advice of the referee we have carefully proofread the document to correct typographical errors.

Since these changes do not have scientific substance we have not included in the resubmission a version highlighting differences.

Many thanks

Tim Rogers, George Constable, Yvonne Krumbeck

PS: Apparently there was some problem with our reply in our first attempt to submit it. We hope it now works. Below is a "point by point" (our replies are lines marked with >>) reply to the requests made to us by the editor and referees.

Associate Editor Comments to Author (Dr Kenta Ishimoto):

Associate Editor: 1

Comments to the Author:

I request of the authors that they should proofread the manuscript and correct typographical errors as suggested by one of the referees.

>> We have carefully proofread the manuscript to correct typographical errors.

Reviewer comments to Author:

Reviewer: 1

Comments to the Author(s)

.

Reviewer: 2

Comments to the Author(s)

The main result of the paper is demonstrating the role of fitness variation in determining the number of mating types emerging in simple evolutionary models. While it is a paper with motivation and interpretation on sexual reproduction, the analysis is based on mathematical models and assumptions. And that requires a very careful balance of a complete, overarching story and technical details (explaining why the reviewers had those issues), which the authors put great effort on before this resubmission.

In general, the authors have seriously responded to those comments and made the flow smoother throughout the manuscript. Unlike the two reviewers before with background on biology, I am a more mathematically minded reader and therefore the mathematical calculations, which are technically sound and beautiful, are within my capabilities to access. On the other hand, it took me much effort to digest the biological justification and arguments (they are significantly supplemented by the authors in this revised version).

Therefore, I would suggest that the editor also refers to the comments from biologists again.

Additionally, the authors may want to double-check the text to eliminate any kind of grammar errors or typos. For instance, in Line 257 and Line 286, it may be more proper to use colons instead of semicolons.

>> Thank you for the positive review. We have carefully proofread the manuscript to correct typographical errors.